# Genetic Variations of the *FUT3* Gene in Le(a−b−) Individuals and Their Association with Lewis Antibody Responses

**DOI:** 10.3390/medsci13040218

**Published:** 2025-10-02

**Authors:** Oytip Nathalang, Piyathida Khumsuk, Wiradee Sasikarn, Kamphon Intharanut

**Affiliations:** 1Graduate Program in Biomedical Sciences, Faculty of Allied Health Sciences, Thammasat University, Pathumtani 12120, Thailand; oytipntl@hotmail.com (O.N.); maimeboice@gmail.com (W.S.); 2Blood Bank, Thammasat University Hospital, Pathumtani 12120, Thailand; piyathida.khu@dome.tu.ac.th

**Keywords:** Lewis blood group, *FUT3*, Lewis antibody, genetic variation, Thai blood donors

## Abstract

**Background:** The biosynthesis of Lewis (Le) antigens depends on the FUT3 gene, encoding an α(1,3/4)-fucosyltransferase. Individuals lacking functional FUT3 exhibit a Le(a–b–) phenotype, regardless of secretor status. **Methods:** This study determined the prevalence of FUT3 single nucleotide variants (SNVs) in Thai blood donors and characterised genotype and allele distributions. We also examined the association between *FUT3* variants and the presence of Le antibodies to better understand variability in immune responses. A total of 112 blood donors were recruited, comprising 52 non-responders and 60 responders for Le antibody detection. The *FUT3* coding sequence was amplified by polymerase chain reaction and directly sequenced to identify single nucleotide variants (SNVs) and haplotypes. **Results:** Associations between *FUT3* SNVs, haplotypes, and Le antibody responsiveness were subsequently analysed. Thirteen *FUT3* SNVs were identified, with c.59T>G (rs28362459) present in all Le(a–b–) cases. The *FUT3*01N.17.03* (*le^59^*^,*1067*^) haplotype was most common (0.634) and showed the strongest association with Le antibody responsiveness (adjusted OR = 3.052, 95% CI: 1.683–5.534, *p* < 0.0001). Differences in antibody types, isotypes, and the *FUT3*01N.17.03* genotype between groups were not statistically significant. **Conclusions:** This first study characterises *FUT3* variations in Le(a–b–) Thai blood donors and identifies *FUT3*01N.17.03* as associated with Le antibody responsiveness, highlighting its relevance for population-specific genetic diagnostics in transfusion medicine.

## 1. Introduction

The International Society of Blood Transfusion (ISBT) Working Party on Red Cell Immunogenetics and Blood Group Terminology designated the Lewis (Le) blood group system ISBT007, which is distinct from other blood group systems in that its glycan antigens are adsorbed onto the red cell membrane from various bodily fluids—plasma—rather than being synthesised intrinsically by erythrocytes [1]. Le antigens are carbohydrate moieties present on fucosylated glycosphingolipids and glycoproteins. They are synthesised by epithelial cells of the digestive, bronchial, mammary, seminal, urinary, and orbital tissues and are secreted either externally via exocrine pathways or into the interstitial space, from which they enter the plasma [2]. Le glycolipids are primarily derived from the epithelium of the digestive tract [3], whereas the glycoprotein-associated forms largely remain in plasma and other body fluids. The system currently comprises six antigens, including five polymorphic antigens (Le^a^, Le^b^, Le^bH^, ALe^b^, and BLe^b^) and one high-prevalence antigen (Le^ab^) [4].

The biosynthesis of Le antigens depends on the *LE* or *FUT3* gene, which encodes an α1,3/4-L-fucosyltransferase (FucT) [5]. The gene is located on chromosome 19q13.3 and comprises three exons spanning approximately 8 kbp of genomic DNA. Its primary product is an enzyme that transfers fucose in an α1→4 linkage to the penultimate N-acetylglucosamine of type 1 carbohydrate precursor chains, which are attached to proteins or lipids on the cell surface. Either the Le^a^ or Le^b^ antigen is formed depending on the sequential functionally activities of the *FUT3* gene and the Secretor gene (*FUT2*), which encodes the α1,2-L-FucT. In secretors, the enzyme catalyses the addition of a L-fucose residue to the Type 1 H antigen to form the Le^b^ antigen; in non-secretors, the enzyme catalyses the addition of the L-fucose residue to the Type 1 precursor to produce the Le^a^ antigen (Figure 1). Consequently, as these structures are acquired from the plasma by the red cell membrane, red cells of most H secretors are Le(a–b+), and those of most H non-secretors are Le(a+b–). The Lewis transferase can also convert Type 1 A to ALe^b^ and Type 1 B to BLe^b^; these are the result of the sequential actions of the *FUT2*, *FUT3*, and *ABO* genes [6]. The weak secretor allele (*Se^w^*), frequently observed in East Asian and Pacific populations, encodes a product with reduced competitive efficiency compared with the functional *Se* allele. This diminished activity favours enhanced Le^a^ synthesis, such that individuals with either the *Se^w^*/*Se^w^* or *Se^w^*/*se* genotype secrete both Le^a^ and Le^b^ and generally exhibit the Le(a+b+) phenotype on red cells [6,7]. Fundamentally, the *FUT3* locus comprises two principal alleles: *Le*, which encodes an active α1,4-L-FucT, and *le*, which lacks functional activity. Individuals homozygous for le fail to produce either Le^a^ or Le^b^ and consequently exhibit the Le(a–b–) red cell phenotype, irrespective of their ABH group or secretor status [6].

The Le^a^ and Le^b^ antigens are the most clinically relevant and are frequently encountered in practice [6]. Four principal Le phenotypes have been described. The reported frequencies of the Le(a+b–) and Le(a–b+) phenotypes are approximately 22% and 72% in Caucasians, 23% and 55% in Black populations, and 0.2% and 73% in Japanese populations, respectively [8]. The Le(a+b+) phenotype, while rare among European and African populations, occurs relatively commonly in East and Southeast Asia, the Pacific region, and Australasia, with a prevalence ranging from 10% to 40% [9,10,11]. In addition, studies in Black populations have demonstrated a comparatively higher frequency of the Le(a–b–) phenotype, with a prevalence of around 22% [6,12]. Three Thai cohorts of blood donors tested with anti-Le^a^ and anti-Le^b^ consistently showed Le(a–b+) as the predominant phenotype (51–66%), followed by Le(a+b–) (11–31%) and Le(a–b–) (18–24%), with little variation across decades. The Le(a+b+) phenotype, however, was not detected in these cohorts [13,14,15].

As with anti-Le^a^, anti-Le^b^ antibodies occur primarily in individuals with the Le(a–b–) red cell phenotype [6]. Although Le antibodies are relatively common, they rarely cause haemolytic transfusion reactions (HTRs), with only a few reports for anti-Le^a^ [16] and even fewer for anti-Le^b^ [17]. This is largely because most are inactive at 37 °C, are neutralised by donor plasma, or are eluted from transfused Le(a–b–) red cells. As a result, they do not accelerate clearance of antigen-positive cells in vivo [18], allowing patients to receive indirect antiglobulin test (IAT)-compatible transfusions, and they rarely induce haemolytic disease of the foetus and newborn (HDFN), since Le antigens are present in foetal secretions but generally absent on foetal red cells [6]. Moreover, Le antibodies function as histocompatibility antibodies in renal transplantation, with Le(a–b–) recipients showing significantly reduced two-year graft survival compared with Le-positive recipients [19,20]. In Thailand, the relatively high prevalence of the Le(a–b–) phenotype contributes to the frequent detection of Le antibodies in both donors and patients [21]. To ensure transfusion safety, national practice mandates the use of Le(a–b–)-compatible blood units for patients who acquire these antibodies.

Allelic variation at the *LE* locus shows marked differences among populations and ethnic groups, and the genotype–phenotype relationship is often complicated [22,23]. The *FUT3* gene spans 8562 bp and contains three exons, with the coding sequence (1086 bp) located entirely within the third exon [1]. Salomaa et al. reported that 90–95% of Le-negative individuals can be identified by screening for four single-nucleotide variants (SNVs)—c.59T>G (rs28362459), c.1067T>A (rs3894326), c.202T>C, and c.314C>T—which are relatively common in the general population [22]. The ISBT has catalogued several SNVs in both *FUT3*-active and -inactive individuals across diverse ethnic groups to date [1]. To our knowledge, no study has yet estimated the distribution of *FUT3* gene SNVs in the Thai population.

Although anti-Le^a^ is not uncommon, it is detected in only about 1 in 300 sera in Denmark and France [24]. Agglutinating alloanti-Le^a^ is confined to individuals with Le(a–b–) red cells, predominantly ABH secretors, and occurs less frequently in group O than in other ABO types [24,25]; unlike anti-Le^a^, these individuals who possessed anti-Le^b^ are generally ABH non-secretors [25]. No study has conclusively shown whether *FUT3* genetic variation contributes to individual differences in anti-Le responsiveness. Certain SNVs may alter the α1,3/4-L-FucT binding region, reducing or abolishing enzyme activity and thereby affecting Le antigen expression. Yet, the wide allelic diversity of *FUT3*, population-specific distributions, and heterogeneous study designs have yielded inconsistent findings [1]. This underscores the need for systematic investigation of *FUT3* polymorphisms and their role in Le antibody formation, particularly in under-studied populations. We aimed to determine the prevalence of *FUT3* SNVs in a Thai blood donor population and to characterise the distribution of *FUT3* genotypes and allele frequencies. In addition, we investigated the relationship between *FUT3* SNVs, alleles, and the presence of Le antibodies in sera to provide further insight into the variability of immune responses to Le antigens.

## 2. Materials and Methods

### 2.1. Study Design, Population, and Data Collection

The study cohort comprised 112 randomly selected, unrelated Thai blood donors exhibiting the Le(a–b–) phenotype. From September to October 2024, 391 first-time donors were recruited from the Blood Bank Unit, Thammasat University Hospital (TUH), following informed consent. Whole peripheral blood samples anticoagulated with ethylenediaminetetraacetic acid (EDTA) were collected from all participants, and serological testing—including Le phenotyping and antibody screening—was subsequently performed. As a result, only 52 Le(a–b–) donors who were non-responders for Le antibodies were included in the study. In addition, 419 donors with positive antibody screens, collected from Fresh Frozen Plasma (FFP) donors between August 2022 and May 2025 at the National Blood Centre, Thai Red Cross Society, Bangkok, Thailand (NBC-TRC), underwent further antibody screening and identification. Of these, 60 FFP donors met the eligibility criteria for the study and were categorised as the responder group. Serum samples were immediately isolated and stored at −20 °C until testing. Demographic variables, including age, sex, and ABO blood group, were obtained from the TUH donor database. The complete study workflow is illustrated in Figure 2.

### 2.2. Lewis Blood Group Phenotyping and Antibody Characterisation

A 5% suspension of red blood cells (RBCs) in Diluent-I (BIO-RAD, Cressier, Switzerland) was prepared and incubated at room temperature (18–25 °C) for 10 min. Microtubes of the ID card “DiaClon Anti-Le^a^” and “DiaClon Anti-Le^b^” (BIO-RAD, Cressier, Switzerland) were appropriately labelled. Ten microlitres of the RBC suspension were dispensed into each corresponding microtube, and the ID card was centrifuged in an ID centrifuge (BIO-RAD, Cressier, Switzerland) for 10 min. Results were interpreted and recorded according to the manufacturer’s instructions.

All donor samples were screened for antibodies using the column agglutination technique (CAT) with Low Ionic Strength Saline (LISS) Coombs and neutral gels on a fully automated ORTHO VISION MAX Analyser (Ortho Clinical Diagnostics, Raritan, NJ, USA) at the NBC-TRC. Samples yielding a positive screening result underwent antibody identification using 11 in-house panel cells (NBC-TRC, Bangkok, Thailand) alongside an auto-control. Antibody specificities were confirmed using both conventional tube testing (CTT).

Briefly, two drops of each plasma sample were mixed with one drop of in-house panel cells (NBC-TRC, Bangkok, Thailand), centrifuged, and examined immediately for agglutination or haemolysis. Reactions were read macroscopically, graded, and recorded. Tubes were then incubated at 37 °C for 30 min, centrifuged, and observed for agglutination. RBCs were washed three times with normal saline, decanted, and two drops of antihuman globulin reagent (CE-Immunodiagnostika GmbH, Neckargemünd, Germany) were added. Agglutination at the IAT phase was graded, following standard guidelines, recorded with weak or negative reactions confirmed microscopically (×10), and validated using IgG-coated RBCs. Reactions at room temperature typically indicate IgM antibodies, whereas those at 37 °C and/or during IAT indicate IgG antibodies.

### 2.3. DNA Extraction

Genomic DNA was isolated from donor whole blood and plasma using QIAamp DNA Blood Kits (Qiagen, Hilden, Germany) in strict accordance with the manufacturer’s protocol. The DNA samples were stored at −20 °C until further analysis.

### 2.4. Polymerase Chain Reaction (PCR) and Direct DNA Sequencing

*FUT3* gene fragments (NG_007482.2 and NM_000149.4), encoding 361 amino acids, were used as references for comparison with reported data and were amplified and sequenced using the primers listed in Table 1. PCR was conducted to target amplify the coding sequence of exon 3 of the *FUT3* gene, using newly designed forward and reverse primers (Table 1). PCR amplification was conducted in a total reaction volume of 40 µL, consisting of 3 µL of genomic DNA (50 ng/µL), 0.75 µM of each forward and reverse primer, 20 µL of 2× PCR Master Mix (i-StarMAXII, iNtRON Biotechnology, Seongnam-Si, Korea), and 11 µL of sterile distilled water. Reactions were performed using a T100 Thermal Cycler (Bio-Rad, Waltham, MA, USA). The target genomic regions were amplified from DNA templates using PCR under the following cycling conditions: an initial denaturation at 95 °C for 1 min; 10 cycles of denaturation at 95 °C for 30 s and annealing at 69 °C for 60 s; followed by 30 cycles of denaturation at 95 °C for 10 s, annealing at 62 °C for 50 s, and extension at 72 °C for 30 s; and a final extension at 72 °C for 5 min. This protocol yielded an 869 bp amplicon (EX3-A1) or a 1042 bp amplicon (EX3-A2).

PCR products were separated on a 1.5% agarose gel containing SYBR™ Safe DNA Gel Stain (Invitrogen, Paisley, UK) and electrophoresed in 1× TBE buffer at 100 Volts. The DNA bands were visualised using a blue-light transilluminator. The desired amplicons were subsequently purified using the GeneJET Gel Extraction Kit (Thermo Scientific, Waltham, MA, USA), and the eluted DNA fragments were sequenced by U2Bio DNA Sequencing Services (Bangkok, Thailand) using the same PCR primers. Alleles for heterozygous mutations were resolved by TOPO TA cloning (Invitrogen, Carlsbad, CA, USA), and ten clones per PCR product were randomly selected for sequencing. Heterozygotes were identified using TOPO cloning of PCR products, followed by Sanger sequencing. Each chromatogram was manually inspected to confirm the presence of both alleles at each variant position.

Linkage disequilibrium (LD) between SNVs was evaluated using haplotype frequencies derived from genotyping data. Specifically, the two variants defining *FUT3*01N.17.03*, c.59T>G and c.1067T>A, were analysed for LD. Strong LD was observed between these sites, with a D′ value of 1. In addition, D, D′, and r^2^ metrics were calculated to quantify the correlation between alleles and to inform haplotype-based association analyses.

### 2.5. Statistical Analysis

All data cleaning and statistical analyses were performed using SPSS, Version 25 (SPSS Inc., Chicago, IL, USA). Phenotype prevalence was summarised using descriptive statistics and reported as percentages. Genotype and allele frequencies were derived by direct counting. Deviations of all SNV genotype frequencies from Hardy–Weinberg equilibrium (HWE) were assessed using a chi-squared goodness-of-fit test. Differences in genotype distributions between Le antibody responders and non-responders were evaluated using chi-square (*χ*^2^) test. Logistic regression models assessing the association between *FUT3* SNVs and Le antibody responsiveness included age and sex as covariates in the final model. Sensitivity analyses incorporating ABO blood group confirmed that the associations remained significant. Associations between *FUT3* SNV alleles and types of antibody responses were further assessed by Fisher’s exact test. Statistical significance was defined as *p* < 0.05 or a 95% confidence interval (CI) excluding unity. The primary endpoint was the association between *FUT3* haplotypes and Le antibody responses. Analyses at the SNV level were considered exploratory. To account for multiple testing across both tiers, *p*-values were adjusted using the Bonferroni correction, yielding an adjusted significance threshold of *p* < 0.006 (0.05/9). For the current post hoc analyses, detectable effect sizes were assessed for the study sample at the Bonferroni-corrected significance threshold and 80% power. Detectable odds ratios were estimated for varying baseline prevalences of Le antibody responses and haplotype frequencies. These analyses were conducted to provide context for the interpretation of association results.

## 3. Results

### 3.1. Participant Characteristics and Responsiveness to Le Antibodies

In total, 112 participants were analysed, of whom 52 were categorised as non-respondent donors and 60 as respondent donors. Their principal characteristics are presented in Table 2. Of the 60 respondent donors, 46 (76.7%) were female and 14 (23.3%) male, with a median age of 28 years (IQR 22–35); 76.7% were aged 21–40 years. Both female representation and this age group were significantly higher than in non-respondents (*p* < 0.05), with a higher female-to-male ratio of 3.29:1, while ABO blood group distribution remained comparable between groups. The existing antibodies, as determined by screening and identification, are summarised in Table 2. In the respondent group, anti-Le^b^ was most frequent (46.6%), followed by anti-Le^a^ (41.7%) and anti-Le^ab^ (11.7%), with IgM antibodies predominating (66.7%), IgM + IgG detected in 25.0%, and IgG alone in 8.3% of these donors.

### 3.2. FUT3 Genetic Variants and Their Association with Le Antigen Responsiveness

PCR amplification of the *FUT3* gene produced the expected fragments of 869 and 1042 bp in all 112 donors. A total of thirteen SNVs were detected within the coding sequence of the *FUT3* gene. Table 3 summarises the genetic and allelic frequencies of the SNVs in the coding sequence. All observed SNV genotype frequencies were consistent with HWE (*p* > 0.05), and call rates exceeded 95%. Notably, the c.59T>G mutation was the most prevalent, occurring in 100.00% of Le(a–b–) cases. The c.1067T>A and c.508G>A variants were observed at frequencies of 87.50% and 35.71%, respectively. By contrast, the variants c.146G>A, c.548C>T, c.612A>G, and c.645T>C were rare, each detected at a frequency of 0.89%. As shown in Table 3, significant differences were observed in the genotype and/or allele frequencies of c.59T>G, c.202T>C, c.314C>T, c.508G>A, and c.1067T>A between respondent and non-respondent groups (*p* < 0.05). However, after adjustment for age and sex using logistic regression models, no genotypes were found to be associated with the Le antibody respondent cohort (*p* > 0.05).

### 3.3. Haplotypes or Alleles and Their Association with Le Antigen Responsiveness

As shown in Table 4, eight haplotypes (alleles) of the *FUT3* gene have been assigned ISBT numbers, while other alleles remain unassigned [1]. Analysis of the *FUT3* coding sequence revealed that the *FUT3*01N.17.03* (*le^59^*^,*1067*^) haplotype was the most prevalent, representing 142 haplotypes (0.634 of the total). This was closely followed by *FUT3*01N.17.05* (*le^59^*^,*508*^), with a frequency of 0.192. Rare haplotypes, including *FUT3*01N.17.13* (*le^59^*^,*548*,*612*,*1067*^) and *FUT3*01N.17.18* (*le^59^*^,*146*,*508*^), were each observed in only a single donor, corresponding to a frequency of 0.004. Moreover, a total of ten novel alleles were discovered in 26 (23.2%) donors and unofficially assigned by ISBT, as summarised in Table 5, arranged according to their occurrence in the respondent and non-respondent cohorts. Among all respondent donors, the *le^59^*^,*796*^, *le^59^*^,*508*,*1067*^, and *le^59^*^,*796*,*1067*^ haplotypes were the most prevalent, with five cases observed for each, accounting for 4.6% of the total respondent haplotypes. In addition, LD analysis between the two SNVs defining *FUT3*01N.17.03*, c.59T>G and c.1067T>A, showed moderate correlation, with D = 0.0245, D′ = 0.717, and r^2^ = 0.078, indicating that these variants are partially linked but not fully predictive of each other.

Multivariable analysis in the final model, adjusted for age and sex, indicated that two alleles—*FUT3*01N.17.03* and *FUT3*01N.17.05*—were significantly associated with Le antibody production (*p* < 0.05). Notably, the most influential allele, based on the highest adjusted odds ratio, was *FUT3*01N.17.03* (aOR = 3.052, 95% CI: 1.683–5.534, *p* < 0.0001), which remained significant after Bonferroni correction (Table 4). Individuals with the Le(a–b–) phenotype carrying the *FUT3*01N.17.03* allele had a 3.052-fold higher likelihood of developing Le-antibodies compared with carriers of other alleles, after controlling for other variables. Post hoc power analysis confirmed that, given the study sample (*n* = 112), a baseline prevalence of approximately 17%, and a haplotype frequency of approximately 63%, this study had 80% power to detect this effect, supporting the robustness of the observed association.

### 3.4. Distribution of Le Antibody Types in FUT3*01N.17.03 Diplotype

All participant donors underwent antibody screening and identification. All 52 non-respondent donors tested negative. Among the 60 respondent donors, the distribution of the *FUT3*01N.17.03* diplotype and corresponding Le antibody types is summarised in Figure 3. Responders were classified according to their predominant type of Le antibody production for analysis. Differences in Le antibody types, their isotypes, and the *FUT3*01N.17.03* genotype between groups were not statistically significant (Fisher exact test: *p* = 0.62 for antibody types, *p* = 0.68 for isotypes). Within respondents, neither anti-Le^a^ antibodies (IgM or IgG) nor the *FUT301N.17.03*/*01N.17.03* genotype showed a similar response compared with anti-Le^b^ antibodies or carriers of *FUT301N.17.03* combined with other alleles.

## 4. Discussion

Irregular erythrocyte antibodies are observed in the general population. Standardised guidelines for the screening of donor blood to detect isoimmune erythrocyte antibodies have been promulgated at the international level and adopted nationally in Thailand. Previous reports indicate that the predominant incidences of erythrocyte alloantibodies are approximately 86% in donor populations and 52% in patient populations when assessed by CTT [21]. These observations are consistent with the relatively high prevalence of the Le(a–b–) phenotype in the Thai population, estimated at 18–24% [13,14,15]. This concordance underscores the clinical relevance of antibody screening in Thailand, where national standards highlight the importance of ensuring the provision of the most compatible blood for transfusion. According to our demographic results, females were significantly over-represented among respondents, with a female-to-male ratio of 3.29:1. This finding is consistent with the observations of Ameen et al. [26], who reported a 0.49% prevalence of alloantibodies in the Kuwaiti population and 2.3% among blood donors, with females exhibiting a threefold higher frequency than males. Similarly, Zhu et al. reported a prevalence of 0.279% among blood donors in the Shaoguan area [27], again noting a higher frequency of irregular erythrocyte alloantibodies in females compared with males. These concordant findings suggest that sex-related immunological factors, particularly sensitisation during pregnancy, may contribute to the increased prevalence of alloantibodies in females. Furthermore, alloantibody frequency was highest among respondent donors aged 21–40 years, significantly exceeding that in non-respondents. This observation is consistent with reports by Makroo et al. [28] and Pahuja et al. [29], highlighting the potential importance of age-related factors in alloantibody formation and the need for targeted donor screening. Our results indicate that ABO/Rh blood type diversity does not significantly influence Le antibody production. Although both Le and ABO antigens are derived from oligosaccharide precursors, the Le system is regulated by fucosyltransferases (*FUT3* and *FUT2*), whereas the ABO system relies on distinct glycosyltransferases. These findings support the view that ABO antigen expression does not impact *FUT3* and therefore does not induce the Le(a–b–) phenotype or subsequent Lewis antibody formation [6].

To further understand the Le antibody response in Le(a–b–) individuals, we assessed all SNVs in the *FUT3* for associations with Le antigen responsiveness. Variants of the *FUT3* gene have been reported in diverse populations, showing ethnic specificity [1,30]. This study discusses the first analysis of *FUT3* variants in Thai individuals with the Le(a–b–) phenotype. Through sequencing and analysis of the coding region, we identified 13 SNVs. All were in HWE with high call rates, indicating a representative population sample and reliable genotyping. Among these, the c.59T>G site was the most prevalent, with a mutation frequency of 100% in Le(a–b–) individuals. This aligns with studies in Northern and Eastern China, which reported frequencies of 43.13% and 39.87% among all Le phenotypes [30,31], suggesting that c.59T>G is a common variant influencing the Le(a–b–) phenotype across Asian populations. Afterwards, c.1067T>A and c.508G>A variants were observed at frequencies of 87.5% and 35.7%, respectively. Together with the other common variants, these findings indicate that these mutations are relatively prevalent and may contribute substantially to the Le(a–b–) phenotype in the population [22]. The rare mutations c.146G>A, c.548C>T, c.612A>G, and c.645T>C, each observed in a single Thai subject, may represent novel or population-specific variants relevant to the Le(a–b–) phenotype. Although the c.1067A and c.508G alleles were associated with Le antibody formation, no genotype differed significantly between responder and non-responder groups after adjusting for age and sex, suggesting additional factors may influence antibody response.

We identified eight ISBT-assigned and ten unassigned haplotypes among the Le(a–b–) Thai donors, with *FUT301N.17.03* (*le^59^*^,*1067*^) and *FUT301N.17.05* (*le^59^*^,*508*^) being the most frequent. This pattern is consistent with reports by Cooling et al. [32] and Pang et al. [33], who noted *le^59^*^,*1067*^ and *le^59^*^,*508*^ as the principal Lewis-negative alleles in Asian and African populations. The *le^59^*^,*508*^ allele is also common in Americans and Japanese but rare in Europeans—Swedish [32,33], indicating population-specific distributions of non-functional *FUT3* variants. Such variation is clinically relevant, as different haplotypes may influence the likelihood and immunogenicity of Le antibody formation, underscoring the need to consider population background in transfusion practice. It is noteworthy that *FUT3*01.10* (*Le^1067^*) and *FUT3*01.17.01* (*Le^59^*) were detected only in six non-respondent donors. These functional *FUT3* variants have been reported to reduce enzyme activity by approximately 90% [1,34,35]. The c.59T>G substitution predominantly decreases enzyme quantity rather than abolishing activity, as it lies within the transmembrane domain and results in a leucine-to-arginine change (p.Leu20Arg) [23]. Importantly, this alteration does not directly affect the catalytic region, which may explain the absence of antibody production in this group. Nevertheless, this observation illustrates that serological phenotyping does not always reflect the underlying genotype. Further genotyping showed that all six individuals carried a functional *FUT2* allele and were therefore secretors. Based on these combined genetic results, their predicted phenotype would be Le(a–b+). As Le antigens are adsorbed onto RBCs, a 90% reduction in their production may render them undetectable by serology.

Our findings highlight that the *FUT3*01N.17.03* haplotype, the most prevalent among Thai donors, is strongly associated with Le antibody responsiveness. Even after adjusting for age and sex in logistic regression models, the association remained significant (aOR = 3.05), indicating that this effect is robust and not confounded by these covariates. Although genotype-level analyses of individual SNVs showed no significant associations, the *FUT3*01N.17.03* haplotype captured their combined effects. Notably, this signal reflects the cumulative impact of its defining SNVs rather than being driven solely by LD, supporting its biological relevance in modulating Le antibody formation. Importantly, this association was not accompanied by heterozygosity differences in either the type or isotype of Le antibodies, suggesting that the underlying genetic mechanism does not alter the qualitative nature of the immune response. Instead, the combined c.59T>G and c.1067T>A substitutions appear to inactivate the α1,3/4-L-FucT enzyme [23], thereby abolishing Le antigen expression on the red cell surface. This lack of antigenic presentation likely serves as the immunological trigger for Le antibody production in otherwise healthy donors. Moreover, the cumulative effect of haplotypes may contribute to the observed statistical significance, highlighting the complex genetic basis of Le antibody formation. These results enhance our understanding of population-level variation in immune responses to Le antigens and suggest practical applications for transfusion medicine. Specifically, the SNVs and haplotypes identified here could guide the refinement of *FUT3* genotyping strategies, including improved SNV filtering and scoring, for the selection of Le-negative phenotypes. In particular, *FUT3* variants with high prevalence and demonstrated correlation with Le antibody formation in the Thai population should be prioritised when designing genetic tests to improve diagnostic accuracy and specificity. Beyond Le phenotyping, *FUT3* variations have been associated with ulcerative colitis [36], ankylosing spondylitis [37], and colonic polyps [38], suggesting that altered Le antigen expression may influence immunity, inflammation, and disease susceptibility. Together, these findings underscore the importance of investigating *FUT3* variants both for understanding Le antibody formation and for their broader clinical relevance, including implications for not only genetic testing but also disease risk assessment.

However, our study has several limitations. First, the sample size is relatively modest, which may lead to wide confidence intervals and reduce the statistical power to detect true differences. Although adjustment for age and sex helped to mitigate potential confounding, it cannot fully eliminate biases arising from non-random sampling. Moreover, for outcomes with low incidence (<10%), OR estimates are inherently unstable and should therefore be interpreted with caution. Second, the Le antibodies identified in this study reflect only the humoral response directed against the LE system, without accounting for possible concomitant antibodies to other blood group antigens. As Le antigens share structural features with several other carbohydrate-based systems, particularly those with terminal oligosaccharide epitopes, cross-reactivity cannot be excluded. The processing of such polysaccharide antigens within endosomes is complex, and only a limited number of highly antigenic domains (e.g., Galili, Forssman, Le^X^/Le^Y^, and P1PK system-antigens) are capable of eliciting strong immune responses, which may contribute to overlapping antibody patterns. Third, additional immune-related genetic factors and health conditions, such as polymorphisms in the *HLA* and *FUT2* genes, which were beyond the scope of the present analysis, may also influence the antibody response to Le antigens.

## 5. Conclusions

To the best of our knowledge, this study provides the first characterisation of *FUT3* gene variants in Thai blood donors with the Le(a–b–) phenotype and identifies *FUT3*01N.17.03* as the key allele associated with Le antibody responsiveness. These findings highlight the population-specific distribution of *FUT3* variants and their potential relevance for genetic diagnostics in transfusion practice. Further studies are warranted to clarify the underlying mechanisms by which *FUT3* SNVs influence antibody formation and to assess their broader clinical implications.

## Figures and Tables

**Figure 1 medsci-13-00218-f001:**
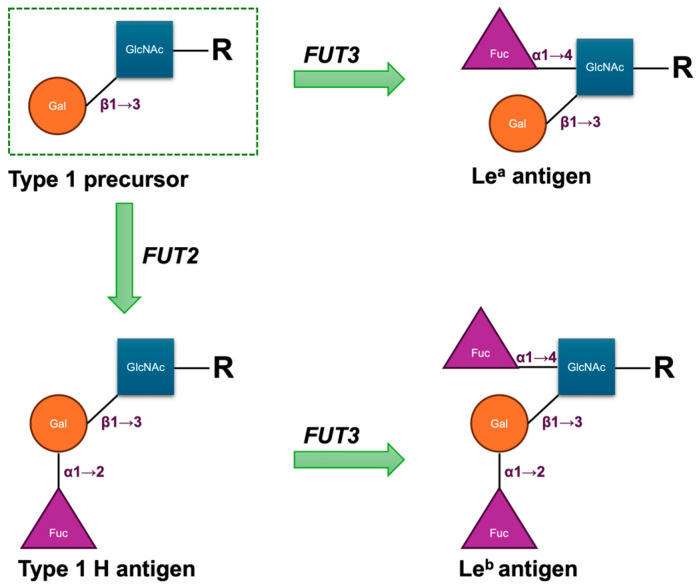
Structural scheme of antigenic domains in Type 1 chain precursors and Lewis antigens. Fuc, fucose; Gal, galactose; GlcNAc, N-acetylglucosamine; R, side chain.

**Figure 2 medsci-13-00218-f002:**
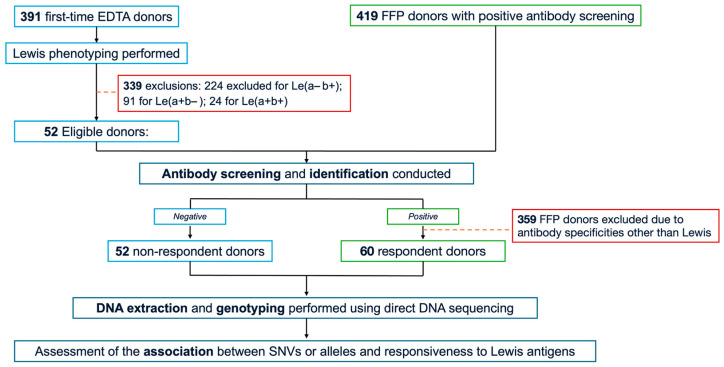
Study flow chart. EDTA, ethylenediaminetetraacetic acid; FFP, fresh-frozen plasma; SNVs, single nucleotide variants.

**Figure 3 medsci-13-00218-f003:**
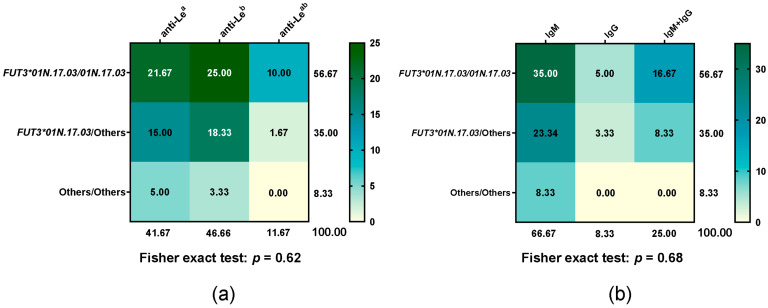
Heatmap of Le antibody type frequencies across *FUT3*01N.17.03* diplotypes, with each cell showing the percentage of that diplotype relative to the total. (**a**) Le antibody types and (**b**) antibody isotypes across *FUT3*01N.17.03* diplotypes. Green indicates higher percentages (with darker shades representing stronger values), while yellow indicates lower percentages.

**Table 1 medsci-13-00218-t001:** Oligonucleotide primers for PCR and sequencing.

Name of Primer	Sequence of Primer (5′ to 3′)	PCR Product Size, bp
LEW-EX3-A1-For	GCAGCTCCTCTCAGGACTCA	869
LEW-EX3-A1-Rev	TGATGTAGTCGGGGTGCAAG	
LEW-EX3-A2-For	CCAAGGGGACCATGATGGAGAC	1042
LEW-EX3-A2-Rev	AGTCGATCCCACCTGTACCCTA	

Abbreviations: bp, base pair.

**Table 2 medsci-13-00218-t002:** Principal demographic characteristics and distribution of Le antibody types among respondent and non-respondent donors.

Variable	Overall (*n* = 112)	Non-Respondent (*n* = 52)	Respondent (*n* = 60)	*χ* ^2^	DF	*p* Value
Sex (F:M ratio)	1.95:1	1.17:1	3.29:1			
Female, n (%)	74 (66.1)	28 (53.8)	46 (76.7)	6.47	1	0.011 *
Male, n (%)	38 (33.9)	24 (46.2)	14 (23.3)			
Age (Year): median (IQR)	28 (22, 39)	30 (24, 42)	28 (22, 35)			
≤20, n (%)	13 (11.6)	7 (13.5)	6 (10.0)	7.16	2	0.028 *
21–40, n (%)	74 (66.1)	28 (53.8)	46 (76.7)			
41–60, n (%)	24 (21.4)	17 (32.7)	8 (13.3)			
ABO group						
A, n (%)	25 (22.3)	7 (13.5)	18 (30.0)	4.59	3	0.204
B, n (%)	45 (40.2)	24 (46.1)	21 (35.0)			
O, n (%)	25 (22.3)	12 (23.1)	13 (21.7)			
AB, n (%)	17 (15.2)	9 (17.3)	8 (13.3)			
Le antibody						
Anti-Le^a^	25 (41.7)	0 (0.0)	25 (41.7)	ND	ND	ND
Anti-Le^b^	28 (46.6)	0 (0.0)	28 (46.6)			
Anti-Le^ab^	7 (11.7)	0 (0.0)	7 (11.7)			
Isotype of antibody						
IgM	40 (66.7)	0 (0.0)	40 (66.7)	ND	ND	ND
IgG	5 (8.3)	0 (0.0)	5 (8.3)			
IgM + IgG	15 (25.0)	0 (0.0)	15 (25.0)			

* Values indicate a *p* value < 0.05. Abbreviations: χ^2^, chi-square; DF, degrees of freedom; F, female; IgG, immunoglobulin G; IgM, immunoglobulin M; IQR, interquartile range; Le, Lewis; M, male; n, number; ND, not determined. Data are median (IQR) or n (%).

**Table 3 medsci-13-00218-t003:** Genotype distribution of *FUT3* SNVs in respondent and non-respondent cohorts.

SNVs (rs Number)	Amino Acid Change	Genotype	Non-Respondent (*n* = 52), *n* (%)	Respondent (*n* = 60), *n* (%)	*p* ^1^	aOR (95% CI) ^2^	*p* ^2^
c.47G>C (rs145362171)	p.Cys16Ser	GG	50 (96.2)	60 (100.0)	0.125	1.00	
		GC	2 (3.8)	0 (0.0)		ND	ND
		CC	0 (0.0)	0 (0.0)		ND	ND
		G	102 (98.1)	120 (100.0)	ND		
		C	2 (1.9)	0 90.0)			
c.59T>G (rs28362459)	p.Leu20Arg	TT	0 (0.0)	0 (0.0)	0.204	1.00	
		TG	6 (11.5)	3 (5.0)		0.354 (0.080–1.558)	0.170
		GG	46 (88.5)	57 (95.0)		2.827 (0.642–12.452)	0.170
		T	6 (5.8)	3 (2.5)	0.014 *		
		G	98 (94.2)	117 (97.5)			
c.146G>A (rs1263565737)	p.Ser49Asn	GG	51 (98.1)	60 (100.0)	0.281	1.00	
		GA	1 (1.9)	0 (0.0)		ND	ND
		AA	0 (0.0)	0 (0.0)		ND	ND
		G	103 (99.0)	120 (100.0)	ND		
		A	1 (0.0)	0 (0.0)			
c.202T>C	p.Trp68Arg	TT	48 (92.3)	58 (96.7)	0.307	1.00	
		TC	4 (7.7)	2 (3.3)		0.354 (0.058–2.162)	0.261
		CC	0 (0.0)	0 (0.0)		ND	ND
		T	100 (96.2)	118 (98.3)	0.029 *		
		C	4 (3.8)	2 (1.7)			
c.314C>T	p.Thr105Met	CC	48 (92.3)	58 (96.7)	0.307	1.00	
		CT	4 (7.7)	2 (3.3)		0.354 (0.058–2.162)	0.261
		TT	0 (0.0)	0 (0.0)		ND	ND
		C	100 (96.2)	118 (98.3)	0.029 *		
		T	4 (3.8)	2 (1.7)			
c.508G>A (rs3745635)	p.Gly170Ser	GG	30 (57.7)	42 (70.0)	0.039 *	1.00	
		GA	17 (32.7)	18 (30.0)		0.837 (0.361–1.941)	0.678
		AA	5 (9.6)	0 (0.0)		ND	ND
		G	77 (74.0)	102 (85.0)	0.001 *		
		A	27 (26.0)	18 (15.0)			
c.548C>T (rs146519599)	p.Pro183Leu	CC	52 (100.0)	59 (98.3)	0.350	1.00	
		CT	0 (0.0)	1 (1.7)		ND	ND
		TT	0 (0.0)	0 (0.0)		ND	ND
		C	104 (100.0)	119 (99.2)	ND		
		T	0 (0.0)	1 (0.8)			
c.612A>G (rs28362465)	p.Ser204=	AA	52 (100.0)	59 (98.3)	0.350	1.00	
		AG	0 (0.0)	1 (1.7)		ND	ND
		GG	0 (0.0)	0 (0.0)		ND	ND
		A	104 (100.0)	119 (99.2)	ND		
		G	0 (0.0)	1 (0.8)			
c.645T>C (rs148170391)	p.Ala215=	TT	51 (98.1)	60 (10.0)	0.281	1.00	
		TC	1 (1.9)	0 (0.0)		ND	ND
		CC	0 (0.0)	0 (0.0)		ND	ND
		T	103 (99.0)	120 (100.0)	ND		
		C	1 (1.0)	0 (0.0)			
c.796G>C	p.Glu266Gln	GG	47 (90.4)	52 (86.7)	0.540	1.00	
		GC	5 (9.6)	8 (13.3)		1.475 (0.425–5.119)	0.541
		CC	0 (0.0)	0 (0.0)		ND	ND
		G	99 (95.2)	112 (93.3)	0.105		
		C	5 (4.8)	8 (6.7)			
c.974C>T (rs28381969)	p.Thr325Met	CC	50 (96.2)	60 (100.0)	0.125	1.00	
		CT	2 (3.8)	0 (0.0)		ND	ND
		TT	0 (0.0)	0 (0.0)		ND	ND
		C	102 (98.1)	120 (100.0)	ND		
		T	2 (1.9)	0 (0.0)			
c.1029A>G (rs199931170)	p.Lys343=	AA	50 (96.2)	60 (100.0)	0.125	1.00	
		AG	2 (3.8)	0 (0.0)		ND	ND
		GG	0 (0.0)	0 (0.0)		ND	ND
		A	102 (98.1)	120 (100.0)	ND		
		G	2 (1.9)	0 (0.0)			
c.1067T>A (rs3894326)	p.Ile356Lys	TT	7 (13.5)	1 (1.7)	0.054	1.00	
		TA	17 (32.7)	22 (36.7)		1.216 (0.537–2.758)	0.639
		AA	28 (53.8)	31 (61.7)		1.462 (0.665–3.212)	0.345
		T	31 (29.8)	24 (20.0)	0.006 *		
		A	73 (70.2)	96 (80.0)			

* Values indicate a *p* value < 0.05. ^1^ Calculated by the chi-square test. ^2^ Adjusted for age and gender by logistic regression models. Abbreviations: aOR, adjusted odd ratio; CI, confidence interval; n, number; ND, not determined; SNVs, single nucleotide variants.

**Table 4 medsci-13-00218-t004:** Haplotype and allele frequencies of the *FUT3* coding sequence in respondent and non-respondent donors.

ISBT Allele	Haplotypes	Number (Frequency)			aOR (95% CI) ^1^	*p* Value ^1^	Adjusted *p* Value ^1^
		Overall (*n* = 224)	Non-Respondent (*n* = 104)	Respondent (*n* = 120)			
*FUT3*01.10*	*Le^1067^*	2 (0.009)	2 (0.019)	0 (0.000)	ND	ND	ND
*FUT3*01.17.01*	*Le^59^*	4 (0.018)	4 (0.038)	0 (0.000)	ND	ND	ND
*FUT3*01N.03.02*	*le^202^* ^,*314*^	4 (0.018)	2 (0.019)	2 (0.017)	0.635 (0.084–4.826)	0.661	0.661
*FUT3*01N.03.11*	*le^47^* ^,*202*,*314*^	1 (0.004)	1 (0.010)	0 (0.000)	ND	ND	ND
*FUT3*01N.17.03*	*le^59^* ^,*1067*^	142 (0.634)	53 (0.510)	89 (0.742)	3.052 (1.683–5.534)	<0.0001	<0.0001 *
*FUT3*01N.17.05*	*le^59^* ^,*508*^	43 (0.192)	27 (0.260)	16 (0.133)	0.396 (0.193–0.812)	0.011	0.011
*FUT3*01N.17.13*	*le^59^* ^,*548*,*612*,*1067*^	1 (0.004)	0 (0.000)	1 (0.008)	ND	ND	ND
*FUT3*01N.17.18*	*le^59^* ^,*146*,*508*^	1 (0.004)	1 (0.010)	0 (0.000)	ND	ND	ND
*Others*	*Others*	26 (0.116)	14 (0.135)	12 (0.100)	0.708 (0.301–1.666)	0.429	0.429

* Values indicate a *p* value < 0.006 according to the Bonferroni correction. ^1^ Adjusted for age and gender by logistic regression models. Abbreviations: aOR, adjusted odd ratio; CI, confidence interval; ISBT, the international society of blood transfusion; ND, not determined.

**Table 5 medsci-13-00218-t005:** Supplementary haplotype frequencies of the *FUT3* coding sequence in respondent and non-respondent donors.

Haplotypes	Overall (*n* = 112)	Non-Respondent (*n* = 52)	Respondent (*n* = 60)
*le^59^* ^,*796*^	5 (0.045)	0 (0.000)	2 (0.033)
*le^59^* ^,*508*,*1067*^	5 (0.045)	3 (0.058)	2 (0.033)
*le^59^* ^,*796*,*1067*^	5 (0.045)	3 (0.058)	2 (0.033)
*le^796^* ^,*1067*^	3 (0.027)	1 (0.019)	2 (0.033)
*le^59^* ^,*974*,*1067*^	2 (0.018)	2 (0.038)	0 (0.000)
*le^59^* ^,*1029*,*1067*^	2 (0.018)	2 (0.038)	0 (0.000)
*le^47^* ^,*202*,*314*,*1067*^	1 (0.009)	1 (0.019)	0 (0.000)
*le^59^* ^,*667*^	1 (0.009)	0 (0.000)	1 (0.017)
*le^59^* ^,*508*,*796*^	1 (0.009)	1 (0.019)	0 (0.000)
*le^59^* ^,*645*,*1067*^	1 (0.009)	1 (0.019)	0 (0.000)
Total	26 (0.232)	14 (0.269)	12 (0.200)

Abbreviations: n, number.

## Data Availability

The raw data supporting the conclusions of this article will be made available by the authors on request.

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
