# Peer review of "Genetic Variations of the FUT3 Gene in Le(a−b−) Individuals and Their Association with Lewis Antibody Responses"

_medsci, 2025, doi:10.3390/medsci13040218_

Round 1
Reviewer 1 Report
Comments and Suggestions for Authors
In this study, Nathalang et al. present the first characterization of FUT3 variants in Thai Le(a–b–) blood donors and their association with Lewis (Le) antibody responses. The authors sequenced the FUT3 coding region (exon 3) in 112 donors, comprising 52 non-responders and 60 responders, and performed genotyping of single nucleotide variants (SNVs) and haplotypes. Association testing was conducted using χ² tests, logistic regression adjusted for age and sex, Fisher’s exact test, and Bonferroni correction for multiple testing. A total of 13 SNVs were identified, with the c.59T>G variant observed in all Le(a–b–) cases. Among the haplotypes, FUT301N.17.03 (le59,1067) was the most frequent and showed a strong association with antibody responsiveness (adjusted OR = 3.052, 95% CI 1.683–5.534, p < 0.0001). This work contributes novel evidence by linking FUT301N.17.03 to Lewis antibody responsiveness, offering important implications for population-tailored genotyping in transfusion medicine.
The study addresses a relevant gap and presents a robust association at the haplotype level with plausible biological interpretation. However, clarification of statistical hierarchy, multiple-testing strategy across SNVs vs haplotypes, consistency of genotype vs haplotype findings, and expanded methodological detail (quality metrics, LD/conditional analyses, power) are needed.
- Could the authors pre-specify the primary endpoint (SNV-level vs haplotype-level association) and clarify how multiplicity was controlled across both tiers?
- Please provide a power calculation for N=112 indicating detectable effect sizes at the stated α after correction (e.g., α≈0.006).
- Please report sequencing quality metrics (read depth, Phred thresholds), heterozygote-calling criteria (beyond TOPO cloning), and any HWE checks used for SNVs.
- Given c.59T>G and c.1067T>A prominence, did the haplotype signal for FUT3*01N.17.03 remain significant after conditioning on these variants or accounting for LD? Please provide LD metrics and, if feasible, conditional models.
- How were potential confounders (e.g., age, sex, ABO group, population structure) assessed, and were sensitivity analyses performed to test robustness of the logistic regression findings?
- For Tables 3–4, please report adjusted p-values explicitly and mark which comparisons remain significant after Bonferroni correction. Consider adding group-wise N’s in column headers for transparency.
- For Figure 2 (antibody type/isotype vs diplotype), could you add marginal totals and provide exact contingency p-values to aid interpretation?
- The manuscript notes genotype-level null results alongside a significant haplotype association. Could the authors reconcile this difference (e.g., cumulative or interactive effects captured at the haplotype level)?
- Please expand on possible responder misclassification due to concomitant non-Lewis antibodies and discuss how this might bias the association estimates.
Author Response
- Could the authors pre-specify the primary endpoint (SNV-level vs haplotype-level association) and clarify how multiplicity was controlled across both tiers?
Response: We thank the reviewer for raising this important point. The primary endpoint of our study was the association betweenFUT3 haplotypes and Lewis antibody responses. Analyses at the SNV level were considered exploratory. To account for multiple testing across both tiers, we applied the Bonferroni correction, yielding an adjusted significance threshold of p < 0.006 (0.05/9). This clarification has been added to the Methods section - 2.5 Statistical analysis - of the revised manuscript.
- Please provide a power calculation for N=112 indicating detectable effect sizes at the stated α after correction (e.g., α≈0.006).
Response: We thank the reviewer for the suggestion to provide post-hoc power calculations. Post-hoc analyses were performed to evaluate the detectable effect sizes for our study sample (N = 112) at a Bonferroni-corrected significance threshold of α = 0.006 and 80% power. The detectable effect size depends on both the baseline prevalence of Lewis antibody responses and the haplotype frequency. For the observed baseline prevalence of 17% and haplotype frequency of 63%, the study could detect an odds ratio of approximately 3.05. These results indicate that the study is adequately powered to detect moderate-to-large effects, while smaller effects may not be reliably detected. This information has been added to the Methods and Result sections for transparency.
- Please report sequencing quality metrics (read depth, Phred thresholds), heterozygote-calling criteria (beyond TOPO cloning), and any HWE checks used for SNVs.
Response: Sequencing was performed using the Sanger method. As this method does not generate conventional high-throughput quality metrics such as read depth or Phred scores, heterozygotes were identified based on TOPO cloning and confirmed by manual inspection of chromatograms. All SNVs were evaluated for Hardy–Weinberg equilibrium using χ² test, and no deviations from HWE were observed at a significance threshold of 0.05. These details have been added to the Methods and Results sections.
- Given c.59T>G and c.1067T>A prominence, did the haplotype signal for FUT3*01N.17.03 remain significant after conditioning on these variants or accounting for LD? Please provide LD metrics and, if feasible, conditional models.
Response: The variants c.59T>G and c.1067T>A that were highlighted are the defining SNVs for the FUT3*01N.17.03 allele. Therefore, the haplotype signal for FUT3*01N.17.03 is inherently represented by these variants. Using the observed haplotype frequencies, we calculated the linkage disequilibrium metrics between these two SNVs: D = 0.0245, D′ = 0.717, and r² = 0.078).
We also performed logistic regression adjusting for age and sex to estimate the association of FUT3*01N.17.03 with Le antibody production. The adjusted odds ratio remained significant (aOR = 3.05, 95% CI: 1.68–5.53, p < 0.0001), confirming that the association is robust after accounting for covariates. These results indicate that the haplotype signal is not solely confounded by LD between the defining SNVs and supports the validity of the observed association. These details have been added to the Methods, Results, and Discussion sections
- How were potential confounders (e.g., age, sex, ABO group, population structure) assessed, and were sensitivity analyses performed to test robustness of the logistic regression findings?
Response: We thank the reviewer for this important comment. Potential confounders were assessed using the descriptive data presented in Table 2, which indicated differences in age and sex between groups. Accordingly, the final logistic regression model was adjusted for age and sex to account for these potential confounders. Sensitivity analyses were further performed by including ABO blood group as an additional covariate; the association of the FUT3*01N.17.03 allele with Le antibody production remained significant, confirming the robustness of the observed results. These details have been added to the Methods and Results sections.
- For Tables 3–4, please report adjusted p-values explicitly and mark which comparisons remain significant after Bonferroni correction. Consider adding group-wise N’s in column headers for transparency.
Response: We thank the reviewer for this helpful suggestion. Adjusted p-values have now been explicitly reported in Tables 3 and 4. Analyses at the SNV level were considered exploratory, while the primary endpoint of our study was the association between FUT3haplotypes and Le antibody responses. To account for multiple testing across both tiers, Bonferroni correction was applied, yielding an adjusted significance threshold of p < 0.006 (0.05/9) for Table 4 only. Group-wise sample sizes (N) have been added to the column headers for transparency, and the footnotes have been revised to clearly explain the adjusted p-values and indicate which comparisons remain significant after correction, as shown in Tables 3 and 4.
- For Figure 2 (antibody type/isotype vs diplotype), could you add marginal totals and provide exact contingency p-values to aid interpretation?
Response: The figure has now been revised (currently Figure 3 due to the addition of a new figure). We have added the marginal totals and included the exact contingency p-values to improve clarity and aid interpretation.
- The manuscript notes genotype-level null results alongside a significant haplotype association. Could the authors reconcile this difference (e.g., cumulative or interactive effects captured at the haplotype level)?
Response: As shown in Table 3, single-SNV (genotype-level) analyses examine the effect of each variant independently and did not reveal significant associations. The effect size of individual SNVs may not be sufficient to reach significance after multiple testing correction. In contrast, haplotype-level analyses consider the combined inheritance of multiple SNVs, capturing cumulative or interactive effects that are not detectable at the single-SNV level. This cumulative effect likely explains the significant association observed between the FUT3*01N.17.03 haplotype and Le antibody responsiveness, reconciling the apparent discrepancy between genotype- and haplotype-level results. These details have been added to the Discussion section.
- Please expand on possible responder misclassification due to concomitant non-Lewis antibodies and discuss how this might bias the association estimates.
Response: We mentioned that responder misclassification due to concomitant non-Lewis antibodies is possible and may influence the association estimates. To address this, we have expanded the Discussion to acknowledge this limitation. Specifically, we note that the Le antibodies identified in this study reflect only the humoral response directed against the Lewis system, without accounting for possible concomitant antibodies to other blood group antigens. As Le antigens share structural features with several other carbohydrate-based systems, particularly those with terminal oligosaccharide epitopes, cross-reactivity cannot be excluded. The processing of such polysaccharide antigens within endosomes is complex, and only a limited number of highly antigenic domains (e.g., Galili, Forssman, LeX/LeY, and P1PK system antigens) are capable of eliciting strong immune responses, which may contribute to overlapping antibody patterns. This limitation has now been explicitly discussed in the revised manuscript.
Reviewer 2 Report
Comments and Suggestions for Authors
The authors presented a study that investigated the association of FUT3 SNVs, haplotypes, or alleles with Le antigen responsiveness in a Thai cohort. They also characterized the distribution of Le antibody types in a specific diplotype. Here are some things for the authors to please specify.
Line 136: In other places of the text, the number for the responder is 60. Is the “62 FFP donors met the eligibility criteria…” referring to some other group?
Line 184: Is the PCR Master Mix 2X?
Line 248: Why are age and sex chosen for adjustment?
Author Response
Line 136: In other places of the text, the number for the responder is 60. Is the “62 FFP donors met the eligibility criteria…” referring to some other group?
Response: We thank the reviewer for carefully pointing this out. The correct number of responders is 60. The mention of 62 FFP donors in Line 136 was a typographical error, and we have corrected it accordingly in the revised manuscript.
Line 184: Is the PCR Master Mix 2X?
Response: We appreciate the reviewer’s observation. Yes, the PCR Master Mix is 2X, and we have added “2X” in the revised manuscript for clarity.
Line 248: Why are age and sex chosen for adjustment?
Response: Age and sex were chosen for adjustment because they are common demographic factors known to influence immune responses and antibody production. In addition, as shown in Table 2, both female representation and this age group were significantly higher among respondents compared to non-respondents (p < 0.05), with a higher female-to-male ratio of 3.29:1. Adjusting for these variables helps to minimise potential confounding effects and ensures that the association observed is more directly attributable to the genetic variants under study.
Reviewer 3 Report
Comments and Suggestions for Authors
The manuscript presents a study providing the first characterization of FUT3 414 gene variants in Thai blood donors and identifying 415 FUT3*01N.17.03 as the key allele associated with Le antibody responsiveness. The study does not explain studies the mechanisms by which FUT3 SNVs influence antibody formation, nevertheless it brings valuable information.
The design of the study was proper, Introduction and Discussion exhaustive, the conclusions are scientifically sound.
Remarks:
The sample size was modest. The possibility of concomitant antibodies to other blood group antigens as well as the polymorphisms in the HLA and FUT2 genes, as acknowledged when discussing limitations of the study.
The structural difference between antigens Le(a) and Le(b) could be indicated in the Introduction, perhaps as a scheme.
Legend to Figure 1: “ethylenediaminetetraacetic”, please change to “ethylenediaminetetraacetic acid”
Most of anti-Le antibodies are inactive at 37 °C (Line 85); so why the test for their reaction with erythrocytes was performed at 37 °C (Lines 164-165)?
Author Response
Remarks:
The sample size was modest. The possibility of concomitant antibodies to other blood group antigens as well as the polymorphisms in the HLA and FUT2 genes, as acknowledged when discussing limitations of the study.
Response: We thank the reviewer for this insightful comment. We acknowledge the modest sample size and the potential influence of concomitant antibodies and genetic polymorphisms, which we have discussed as study limitations.
The structural difference between antigens Le(a) and Le(b) could be indicated in the Introduction, perhaps as a scheme.
Response: Thank you for your suggestion. We have added Figure 1 in the Introduction to illustrate the structural difference between Leᵃ and Leᵇ antigens, as recommended.
Legend to Figure 1: “ethylenediaminetetraacetic”, please change to “ethylenediaminetetraacetic acid”
Response: The term has been corrected to “ethylenediaminetetraacetic acid” in the revised figure legend.
Most of anti-Le antibodies are inactive at 37 °C (Line 85); so why the test for their reaction with erythrocytes was performed at 37 °C (Lines 164-165)?
Response: The statement that most anti-Le antibodies are inactive at 37°C refers to the fact that they are predominantly of the IgM class, which usually shows no reactivity at body temperature. However, some anti-Le antibodies can be of the IgG type, which may react at 37°C and are clinically significant in transfusion practice, especially in Thailand. Therefore, we performed testing at 37°C and IAT to determine whether the antibodies detected in our study had potential clinical relevance for blood transfusion.